# Can Hybrid Geometric Scattering Networks Help Solve the Maximum Clique Problem?

**Yimeng Min**[*]
Department of Computer Science
Cornell University
Ithaca, NY, USA
min@cs.cornell.edu

**Frederik Wenkel**[*]
Department of Mathematics and Statistics
Université de Montréal
Mila – Quebec AI Institute
Montreal, QC, Canada
frederik.wenkel@umontreal.ca

**Michael A. Perlmutter**
Department of Mathematics
University of California
Los Angeles, CA, USA
perlmutter@ucla.edu

**Guy Wolf**
Department of Mathematics and Statistics
Université de Montréal
Mila – Quebec AI Institute
Montreal, QC, Canada
guy.wolf@umontreal.ca

## Abstract

We propose a geometric scattering-based graph neural network (GNN) for approximating solutions of the NP-hard maximum clique (MC) problem. We construct a loss function with two terms, one which encourages the network to find highly connected nodes and the other which acts as a surrogate for the constraint that the nodes form a clique. We then use this loss to train an efficient GNN architecture that outputs a vector representing the probability for each node to be part of the MC and apply a rule-based decoder to make our final prediction. The incorporation of the scattering transform alleviates the so-called oversmoothing problem that is often encountered in GNNs and would degrade the performance of our proposed setup. Our empirical results demonstrate that our method outperforms representative GNN baselines in terms of solution accuracy and inference speed as well as conventional solvers like Gurobi with limited time budgets. Furthermore, our scattering model is very parameter efficient with only $\sim 0.1\%$ of the number of parameters compared to previous GNN baseline models.

## 1 Introduction

The success of Graph Neural Networks (GNNs) for a variety of machine learning tasks [Scarselli et al., 2008, Gori et al., 2005, Kipf and Welling, 2016, Gilmer et al., 2017] has sparked interests in using GNNs to solve graph combinatorial optimization (CO) problems [Li et al., 2018a, Karalias and Loukas, 2020]. For example, Joshi et al. [2019] introduces a GNN-based method for approximately solving the travelling salesman problem (TSP), while Karalias and Loukas [2020] approximate solutions of the maximum clique (MC) problem. Solving such problems in an end-to-end fashion via GNNs is challenging for several reasons. First of all, many CO problems are provably either NP-hard or NP-complete. Therefore, learning CO solvers in a supervised or semi-supervised fashion can be computationally infeasible since the cost of generating the ground truth labels grows exponentially with the problem size. Second, the solution to the CO problems must satisfy a number of constraints

---

[*]Equal contribution; order determined alphabetically

36th Conference on Neural Information Processing Systems (NeurIPS 2022).

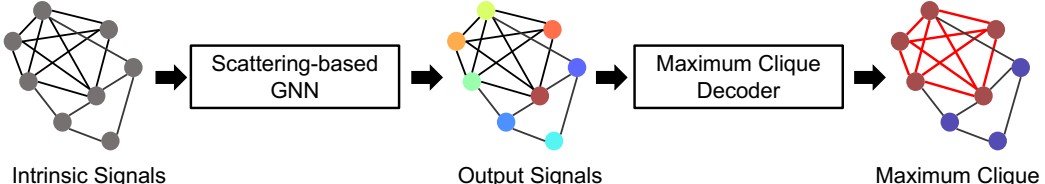

Figure 1: Illustration of our model pipeline. We use a scattering-based GNN to learn a discriminative node representation and use a decoder to extract the MC from the learned representation.

or limitations (e.g. belonging to a clique). Although these constraints can be imposed during training time, there is still no guarantee that these constraints are still satisfied at test time. Furthermore, solving graph combinatorial problems largely depends on the expressive power of GNNs. Many GNNs aggregate information via local averaging, which can be interpreted as a smoothing operation. This degrades the expressive power of GNNs and results in the so-called oversmoothing problem [Li et al., 2018b] where neighboring nodes have similar representations and are difficult to distinguish from each other. This is problematic if a node that is not in the solution set borders many points that are in the solution set.

The purpose of this paper is to use a geometric scattering-based GNN to fast approximate the solution of the maximum clique (MC) problem, that is to find the largest complete subgraph contained within a large graph $G$. This problem has many applications including pattern recognition and knowledge discovery. For instance, finding the maximum clique of the related connection graphs can be used to handle object recognition tasks [Horaud and Skordas, 1989]. It is also tightly related to biological applications, such as helping to solve genome-scale elucidation of biological networks [Zhang et al., 2005]. Our use of geometric scattering, rather than a more traditional GNN is motivated by recent work [Wenkel et al., 2022] showing that the geometric scattering transform can help overcome the oversmoothing problem via the use of band-pass wavelet filters in conjunction with GCN-type filters. This is particularly important in the context of the MC problem because it is critical to distinguish a point which connects to many members of the clique from an actual member of the clique. As we shall show, our method, which utilizes band-pass wavelet filters is able to better detect the border between the MC and the rest of the graph. This parallels the traditional use of wavelets as edge detectors in image processing [Grossmann, 1988].

Our method uses a two-phase strategy shown in Figure 1. We first use an unsupervised learning model to generate an efficient representation of how likely each node is to be part of the maximum clique. Then, this representations is fed into a constraint-preserving decoder to generate the maximum clique. Ideally, given a graph $G$ with $n$ nodes, we would build a representation $\mathbf{p} \in [0, 1]^n$ where the nodes in the maximum clique are assigned the value 1 (True) while all other nodes are assigned 0 (False), so that we can 'cut' the MC from the graph via $O(n)$ operations. In practice, we will build a representation which assigns large probabilities to the maximum clique nodes and small probabilities to the other nodes. This will allow us to extract the maximum clique by choosing the vertices where $\mathbf{p}$

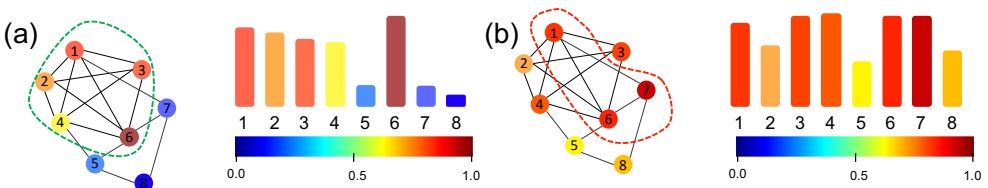

Figure 2: Illustration of the oversmoothing problem. Left: A discriminative representation, which can be interpreted as a probability for each node to belong to the maximum clique or not. Right: Due to the oversmoothing problem, the GNN generates a smooth representation, making MC and non-MC nodes difficult to discriminate.

is large. As illustrated in Figure 2, this approach works best if $\mathbf{p}$ is not excessively smooth. However, so-called oversmoothing is a known limitation of many GNN models [Li et al., 2018b]. Figure 2 (a) presents a situation where oversmoothing does not occur. The signal $\mathbf{p}$ is much larger on the nodes $(1, 2, 3, 4, 6)$, which form the maximum clique, than it is on the other nodes $(5, 7, 8)$. Therefore, the decoder can easily find the MC by, e.g., setting a threshold. Figure 2 (b), on the other hand, illustrates a situation in which oversmoothing occurs and $\mathbf{p}$ exhibits less variability. Indeed, there are no blue colored nodes at all. In this setting, it is nearly impossible to distinguish node 7, which does not belong to the maximum clique, from the nodes $1, 6$, and therefore extracting the MC from $\mathbf{p}$ is difficult if not impossible.

## 2 Background and Related Work

### 2.1 Graph Signal Processing

Consider a graph $G = (V, E)$ with a set of nodes (or vertices) $V := \{v_1, \ldots, v_n\}$ and a set of undirected edges $E \subset V \times V$. For a function $x : V \to \mathbb{R}$ defined on $V$, we will, in minor abuse of notation, identify $x$ with the vector $\mathbf{x}$, where $\mathbf{x}[i] = x(v_i)$. We let $\mathbf{W} \in \mathbb{R}^{n \times n}$ denote the *adjacency matrix* of $G$ and define the *symmetric normalized graph Laplacian* by $\mathcal{L} := \mathbf{I}_n - \mathbf{D}^{-1/2} \mathbf{W} \mathbf{D}^{1/2}$, where $d_i := \sum_{j=1}^{n} W[v_i, v_j]$ is the *degree* of the node $v_i$ and $\mathbf{D} := \operatorname{diag}(d_1, ..., d_n) \in \mathbb{R}^{n \times n}$ is the degree matrix. We will let $\mathbf{q}_i$ and $\lambda_i$ denote the (normalized) eigenvectors and eigenvalues of $\mathcal{L}$ with $\mathcal{L} \mathbf{q}_i = \lambda_i \mathbf{q}_i$, and make use of the eigendecomposition $\mathcal{L} = \mathbf{Q} \mathbf{\Lambda} \mathbf{Q}^\top$, where $\mathbf{Q}$ is the orthogonal matrix whose $i$-th column is the normalized eigenvector $\mathbf{q}_i$, and $\mathbf{\Lambda} := \operatorname{diag}(\lambda_1, ..., \lambda_n)$. Notably, the eigenvectors $\mathbf{q}_1, \ldots \mathbf{q}_n$ can be seen as a generalization of Fourier modes in the context of graph domains with the corresponding eigenvalues representing the increasing frequencies $0 \leq \lambda_1 \leq \cdots \leq \lambda_n \leq 2$. This link may be established by viewing frequency as a measure of variation, in particular the variation of (degree-normalized) Fourier modes across edges, i.e.,

$$\lambda_i = \mathbf{q}_i^\top \mathcal{L} \mathbf{q}_i = \sum_{\{u,v\} \in E} (\tilde{\mathbf{q}}_i[u] - \tilde{\mathbf{q}}_i[v])^2,$$

for $\tilde{\mathbf{q}}_i := \mathbf{D}^{-1/2} \mathbf{q}_i$. The graph *Fourier transform* of a signal vector $\mathbf{x}$ is then defined as by $\hat{\mathbf{x}}[i] = \langle \mathbf{x}, \mathbf{q}_i \rangle$ and the inverse Fourier is given by $\mathbf{x} = \sum_{i=1}^{n} \hat{\mathbf{x}}[i] \mathbf{q}_i$. Compactly written, we have $\hat{\boldsymbol{x}} = \boldsymbol{Q}^\top \boldsymbol{x}$ and $\boldsymbol{x} = \boldsymbol{Q} \hat{\boldsymbol{x}}$.

In the Euclidean setting, it is known that convolution in the spatial domain corresponds to multiplication in the Fourier domain. This fact has motivated work such as Shuman et al. [2016], which defines the convolution of a signal $\mathbf{x}$ with a filter $\mathbf{g}$ to be the unique vector verifying $(\widehat{\boldsymbol{g} \star \mathbf{x}})[i] = \hat{\boldsymbol{g}}[i] \hat{\boldsymbol{x}}[i]$, which implies that

$$\mathbf{g} \star \mathbf{x} = \sum_{i=1}^{n} \hat{\boldsymbol{g}}[i] \hat{\boldsymbol{x}}[i] \mathbf{q}_i = \sum_{i=1}^{n} \hat{\boldsymbol{g}}[i] \langle \mathbf{q}_i, \mathbf{x} \rangle \mathbf{q}_i = \mathbf{Q} \widehat{\boldsymbol{G}} \mathbf{Q}^\top \mathbf{x},$$

where $\widehat{\boldsymbol{G}} := \operatorname{diag}(\hat{\boldsymbol{g}}) = \operatorname{diag}(\hat{\boldsymbol{g}}[1], \ldots, \hat{\boldsymbol{g}}[n])$. When incorporating this notion of convolution into a graph neural network, a common choice [e.g., Defferrard et al., 2016] is to require the coefficients $\hat{\boldsymbol{g}}[i]$ to be polynomials of the eigenvalues $\lambda_i, i \in [n]$, i.e., $\hat{\boldsymbol{g}}[i] := \sum_k \gamma_k \lambda_i^k$ in which case we have $\widehat{\boldsymbol{G}} = \sum_k \gamma_k \mathbf{\Lambda}^k$. This allows one to implement convolution in the spatial domain by verifying that $\boldsymbol{g} \star \boldsymbol{x} = \sum_k \gamma_k \mathcal{L}^k \boldsymbol{x}$. In particular, the entire method may be implemented without the need to diagonalize a matrix which is extremely expensive for large graphs.

### 2.2 Graph Convolutional Networks

In Kipf and Welling [2016], the authors set $\hat{\boldsymbol{g}}[i] := \theta(2 - \lambda_i)$, where $\theta$ is a real number, which yields

$$\mathbf{g}_\theta \star \mathbf{x} = \theta(2 \mathbf{I}_n - \mathcal{L}) = \theta \left( \mathbf{I}_n + \mathbf{D}^{-1/2} \mathbf{W} \mathbf{D}^{-1/2} \right) \mathbf{x}. \tag{1}$$

As the eigenvalues of the above convolutional filter lie in [0,2] and could lead to vanishing or exploding gradients, the authors then apply a renormalization trick which replaces $\mathbf{I}_n + \mathbf{D}^{-1/2} \mathbf{W} \mathbf{D}^{-1/2}$ with

$\mathbf{A} := (\mathbf{D} + \mathbf{I})^{-1/2} (\mathbf{W} + \mathbf{I})(\mathbf{D} + \mathbf{I})^{-1/2}$. The layer-wise propagation rule of GCN is then defined by

$$\mathbf{x}_j^{\ell} = \sigma \Big( \sum_{i=1}^{N_{\ell-1}} \theta_{ij}^{\ell} \, \mathbf{A} \, \mathbf{x}_i^{\ell-1} \Big),$$

where $\mathbf{x}_i^{\ell-1} \in \mathbb{R}^n$ is the $i$-th feature vector in layer $\ell - 1$, $\theta_{ij}^{\ell}$ are a trainable parameters, $\mathbf{x}_j^{\ell}$ is the $j$-th feature activation vector in layer $\ell$ and $\sigma(\cdot)$ is a nonlinear activation function. We can summarize this in matrix notation as

$$\mathbf{X}^{\ell} = \sigma \left( \mathbf{A} \, \mathbf{X}^{\ell-1} \, \boldsymbol{\Theta}^{\ell} \right), \tag{2}$$

where $\boldsymbol{\Theta}^{\ell} \in \mathbb{R}^{N_{\ell-1} \times N_{\ell}}$ is the weight-matrix of layer $\ell$ and $\mathbf{X}^{\ell} \in \mathbb{R}^{n \times N_{\ell}}$ contains the activations output by layer $\ell$. The GCN model updates node features at every node by averaging over the node features of the node itself and its neighbors, which enforces similarity throughout node neighborhoods. This makes nodes increasingly difficult to discriminate for 'deeper' models and is widely referred to as the oversmoothing problem [Li et al., 2018b]. From a graph spectral theory perspective, oversmoothing is related to low-pass filtering as the filter in Eq. 1 puts larger weights on the low-frequency spectrum (as $0 \leq \lambda_i \leq 2$). This motivates the idea of using GCN-type low-pass filters in conjunction with band-pass filters, that can be implemented, for example, using graph scattering [Min et al., 2020], which we discuss in the following.

## 2.3  Graph Scattering

The graph scattering transform [Gama et al., 2019, Zou and Lerman, 2019, Gao et al., 2019] is a wavelet-based model for machine learning on graphs. Unlike the one-hop localized low-pass filters used in GCN, which aim to promote smoothness between neighboring nodes, these wavelets are band-pass filters, which in turn incorporate long-range dependencies through the large spatial support of the used aggregations. The scattering transform is based upon raising the lazy random walk matrix

$$\mathbf{P} := \frac{1}{2} \left( \mathbf{I}_n + \mathbf{W} \, \mathbf{D}^{-1} \right)$$

to different powers in order to capture the diffusion geometry of the graph $G$ at various time scales. In particular, subtracting such powers allows us to detect changes in these diffusion geometries. Following the lead of Coifman and Maggioni [2006], for $k \in \mathbb{N}_0$, we define a wavelet matrix $\boldsymbol{\Psi}_k \in \mathbb{R}^{n \times n}$ at scale $2^k$ by

$$\boldsymbol{\Psi}_0 := \mathbf{I}_n - \mathbf{P}, \quad \boldsymbol{\Psi}_k := \mathbf{P}^{2^{k-1}} - \mathbf{P}^{2^k}, \quad k \geq 1. \tag{3}$$

Intuitively, at every node, these diffusion wavelets can be seen as a comparison operation that calculates the difference of the averaged features of two neighborhoods of different sizes (namely sizes $2^{k-1}$ and $2^k$). The geometric scattering transform is constructed through an alternating cascade on wavelet filters and pointwise nonlinearities $\sigma : \mathbb{R} \to \mathbb{R}$, a so-called scattering path $p := (k_1, \ldots, k_m)$ that outputs

$$\boldsymbol{U}_p \boldsymbol{x} := \boldsymbol{\Psi}_{k_m} \circ \sigma \circ \boldsymbol{\Psi}_{k_{m-1}} \cdots \sigma \circ \boldsymbol{\Psi}_{k_2} \circ \sigma \circ \boldsymbol{\Psi}_{k_1} \boldsymbol{x}. \tag{4}$$

## 3  Model

Our method is centered around three main components, (i) a hybrid scattering-GCN model $M$ (Sec. 3.1-3.3) that transforms a small set of simple node-level statistics represented by the matrix $\mathbf{X} \in \mathbb{R}^{n \times d}$ to a probability vector $\mathbf{p} \in [0, 1]^n$ representing the probabilities that each node is part of the maximum clique, (ii) an easy-to-optimize unsupervised loss function $L$ (Sec. 3.4) that is small for "good" probability vectors $\mathbf{p}^{\star}$; and (iii) a rule-based decoder $D$ (Sec. 3.5) that maps the probability vector to the maximum clique $C^{\star} \subset V$ of $G$.

$$\mathbf{X} \xrightarrow{M} \mathbf{p} \xleftrightarrow{L} \mathbf{p}^{\star} \xrightarrow{D} C^{\star}$$

## 3.1  Embedding Module

The input of our model is a node feature matrix $\mathbf{X} \in \mathbb{R}^{n \times d}$, containing $d$ features for each node. In our experiments, we set $d = 3$ and let the features be the eccentricity, the clustering coefficient, and

the logarithm of the degree of each node. The encoder transforms the node features to $d_h$-dimensional embeddings $\mathbf{H}^0$ using a multi-layer perceptron (MLP) $\mathrm{m_{emb}} : \mathbb{R}^d \to \mathbb{R}^{d_h}$, i.e., $\mathbf{H}^0 := \mathrm{m_{emb}}(\mathbf{X})$, which will be used as the input to the diffusion module introduced in the next subsection.

## 3.2 Diffusion Module

The diffusion module consists of a cascade of $K \in \mathbb{N}$ aggregation (or diffusion) layers with operations that are chosen for each node via an attention mechanism. Our network is based upon the framework introduced in Wenkel et al. [2022]. However, an important deviation is to store the initial node representation $\mathbf{H}^0$ and every intermediate node representation $\mathbf{H}^\ell$, $1 \leq \ell \leq K$, in a list of *readouts*.

In each layer $\ell$, every node has access to node representations from a set a of filters $\mathcal{F}$ that contains a selection of low-pass and band-pass filters. Similar to Wenkel et al. [2022], our band-pass filters $f_{\mathrm{low},r}$ are modified GCN filters that have the form

$$f_{\mathrm{low},r}(\mathbf{H}^{\ell-1}) = \mathbf{A}^r \, \mathbf{H}^{\ell-1} \tag{5}$$

and are parameterized by the power $r \geq 1$ of the matrix $\mathbf{A}$. Similarly, we define the scattering filter $f_{\mathrm{band},k}$ of order $k$ according to

$$f_{\mathrm{band},k}(\mathbf{H}^{\ell-1}) = \mathbf{\Psi}_k \, \mathbf{H}^{\ell-1} \,. \tag{6}$$

Next, we want to obtain data-driven scores $s_f(v)$ that determine the importance of filter $f \in \mathcal{F}$ for node $v \in V$. We set $\mathbf{H}_f^\ell := f(\mathbf{H}^{\ell-1})$ and calculate the scores via an attention mechanism, setting

$$\mathbf{s}_f^\ell := \sigma \left( \mathbf{H}_f^\ell \,\|\, \mathbf{H}^{\ell-1} \right) \mathbf{a}^\ell, \tag{7}$$

where $\|$ denotes the concatenation operation. The attention mechanism, at every node, calculates the dot product of a learned attention vector $\mathbf{a}^\ell \in \mathbb{R}^{2d_h}$ with the filtered node representation of the node concatenated to its previous representation. We further deviate from the model in Wenkel et al. [2022] by rewiring the nonlinearity $\sigma(\cdot)$ in Eq. 7. They apply it after the multiplication with the attention vector $\mathbf{a}^\ell$ similar to the traditional graph attention mechanism [Veličković et al., 2018]. In contrast, we apply it before the multiplication with the attention vector following the lead of Brody et al. [2021], who showed this constitutes an even more expressive attention mechanism. The $i$-th entry of $\mathbf{s}_f^\ell \in \mathbb{R}^n$ then contains the importance score of filter $f$ for node $v_i$. We normalize the scores across the filters using the softmax function, i.e., $\alpha_f(v) = \mathrm{softmax}_{\mathcal{F}}(s_f(v))$, and store the resulting scores in attention vectors $\boldsymbol{\alpha}_f^\ell \in \mathbb{R}^n$, $f \in \mathcal{F}$. We then update node representations via

$$\mathbf{H}_{\mathrm{agg}}^\ell := \sum_{f \in \mathcal{F}} \boldsymbol{\alpha}_f^\ell \odot \mathbf{H}_f^\ell \,. \tag{8}$$

where $\odot$ is the element-wise multiplication (applied separately to each column of $\mathbf{H}_f^\ell$). Notably, the softmax function enforces that every node is likely to focus mostly on one single filter. We note that the method is compatible with multi-head attention but omit further details as it is not necessary for the present experiments. Finally, we transform the aggregated node representations via an MLP $\mathrm{m}^\ell : \mathbb{R}^{d_h} \to \mathbb{R}^{d_h}$, i.e., $\mathbf{H}^\ell := \mathrm{m}^\ell(\mathbf{H}_{\mathrm{agg}}^\ell)$. We repeat the above process for $K \in \mathbb{N}$ iterations (or layers) and store the intermediate representations in a list of readouts $\mathcal{R} = \{\mathbf{H}^\ell\}_{\ell=0}^K$.

## 3.3 Output Module

In the output module, we combine the information from the list of readouts $\mathcal{R}$. We first concatenate the readouts horizontally to

$$\mathbf{H}_{\mathrm{cat}} := \|_{\ell=0}^K \mathbf{H}^\ell,$$

where $\mathbf{H}_{\mathrm{cat}} \in \mathbb{R}^{n \times d_h(K+1)}$ and then transform $\mathbf{H}_{\mathrm{cat}}$ to a vector $\mathbf{h} \in \mathbb{R}^n$ using an MLP $\mathrm{m_{out}} : \mathbb{R}^{d_h(K+1)} \to \mathbb{R}$, i.e., $\mathbf{h} := \mathrm{m_{out}}(\mathbf{H}_{\mathrm{cat}})$. Finally, to obtain a vector of probabilities, we apply min-max normalization, i.e., $\mathbf{p} := (\mathbf{h} - \min(\mathbf{h}) \cdot \mathbf{1}_n)/(\max(\mathbf{h}) - \min(\mathbf{h}))$, and interpret the $i$-th entry of $\mathbf{p}$ as the probability that node $v_i$ is part of the maximum clique.

## 3.4 Training Loss

We now derive an unsupervised training loss for the maximum clique problem, which was also obtained by a different derivation in the work of Karalias and Loukas [2020]. Letting $\Omega$ denote the set of cliques of a graph $G = (V, E)$, we note that the task of finding the maximum clique is equivalent to identifying the clique $C \in \Omega$ that contains the most edges, i.e., maximizes the objective function

$$L^\star(C) := \sum_{u,v \in C} w_{u,v},$$

where $w_{u,v} = 1$ if node $u$ and $v$ are connected otherwise 0. Our approach relies on producing a vector $\mathbf{p}$ such that $\mathbf{p}_v \approx 1$ if $v$ is in the maximum clique and $\mathbf{p}_v \approx 0$ otherwise. We consider the vector $\mathbf{x} \in \mathbb{R}^n$ where $\mathbf{x}_v = 1$ if $v$ is in the maximum clique and $\mathbf{x}_v = 0$ otherwise. We then model $\mathbf{x}_v$ as a Bernoulli random variable where $P(\mathbf{x}_v = 1) = \mathbf{p}_v$. Our goal then becomes to maximize the expectation

$$\mathbb{E}\left[L^\star(C)\right] = \mathbb{E}\left[\sum_{(u,v)\in E} w_{u,v}\, \mathbf{x}_u\, \mathbf{x}_v\right] = \sum_{(u,v)\in E} w_{u,v}\, \mathbf{p}_u\, \mathbf{p}_v = \mathbf{p}^\top \mathbf{W}\, \mathbf{p}. \tag{9}$$

Maximizing Eq. 9 encourages $\mathbf{p}$ to be large on highly connected nodes. However, we also want the support of $\mathbf{p}$ to be concentrated within a clique. This naturally motivates the constrained optimization problem of minimizing

$$L_1(\mathbf{p}) := -\mathbf{p}^\top \mathbf{W}\, \mathbf{p} \quad \text{subject to: } \operatorname{supp}(\mathbf{p}) \text{ is contained in a clique.}$$

However, the constraint that $\mathbf{p}$ is contained in a clique is hard to enforce directly. In order to construct a surrogate, we consider the complement graph $\overline{G} = (\overline{V}, \overline{E})$. The vertices of $\overline{G}$ are taken to be the same as $G$, i.e., $\overline{V} = V$, while the edges are defined by the rule that for $u \neq v$ there is an edge between $u$ and $v$ in $\overline{G}$ if and only if there is *not* an edge between them in $G$, i.e., $\overline{E} = \{(u,v) \in V \times V : u \neq v\}\backslash E$. We denote the adjacency matrix of $\overline{G}$ by $\overline{\mathbf{W}} = (\overline{w}_{u,v})_{u,v\in V}$, and note that

$$\overline{w}_{u,v} := \begin{cases} 1 & \text{if } (u,v) \notin E \text{ and } u \neq v, \\ 0 & \text{otherwise.} \end{cases}$$

In particular, $\overline{\mathbf{W}}$ has zeros on the diagonal and can be computed from $\mathbf{W}$ via $\overline{\mathbf{W}} = \mathbf{1}_{n\times n} - (\mathbf{I} + \mathbf{W})$. The following Lemma indicates that a second loss term $L_2(\mathbf{p}) := \mathbf{p}^\top \overline{\mathbf{W}}\, \mathbf{p}$ can be used to ensure that mass is primarily concentrated in a set of nodes that form a clique.

**Lemma 1.** *Consider a graph $G = (V, E)$, a signal $\mathbf{p} \geq 0$ and define $\operatorname{supp}(\mathbf{p}) := \{v \in V : \mathbf{p}_v > 0\}$. Then, $L_2(\mathbf{p}) = \mathbf{p}^\top \overline{\mathbf{W}}\, \mathbf{p} = 0$ if and only if there exists a clique $C \subset V$ such that the support of $\mathbf{p}$ is contained in $C$.*

*Proof.* We first assume $\mathbf{p}^\top \overline{\mathbf{W}}\, \mathbf{p} = 0$ and suppose for the sake of a contradiction that there exists no clique $C \subset V$ with $\operatorname{supp}(\mathbf{p}) \subset C$. Then, there exist at least two nodes $u_0, v_0 \in \operatorname{supp}(\mathbf{p})$ with $\{u_0, v_0\} \notin E$. Now, $\mathbf{p}^\top \overline{\mathbf{W}}\, \mathbf{p} = \sum_{\{u,v\}\notin E} \mathbf{p}_u\, \mathbf{p}_v \geq \mathbf{p}_{u_0}\, \mathbf{p}_{v_0} > 0$, which is a contradiction.

For the opposite direction, we assume that there exists a clique $C \subset V$ such that $\operatorname{supp}(\mathbf{p}) \subset C$. Hence, every pair of distinct nodes in $\operatorname{supp}(\mathbf{p})$ is connected by an edge and consequently $\overline{w}_{u,v} = 0$ for all $u, v \in \operatorname{supp}(\mathbf{p})$. Thus, $\mathbf{p}^\top \overline{\mathbf{W}}\, \mathbf{p} = \sum_{u,v\in V} \overline{w}_{u,v}\, \mathbf{p}_u\, \mathbf{p}_v = \sum_{u,v\in\operatorname{supp}(\mathbf{p})} \overline{w}_{u,v}\, \mathbf{p}_u\, \mathbf{p}_v = 0$. $\square$

In light of Eq. 9 and Lemma 1, we now define our training loss as

$$L(\mathbf{p}) := L_1(\mathbf{p}) + \beta L_2(\mathbf{p}) = -\mathbf{p}^T \mathbf{W}\, \mathbf{p} + \beta\, \mathbf{p}^T \overline{\mathbf{W}}\, \mathbf{p}. \tag{10}$$

We note that the loss function decomposes into two terms. The first term, $L_1$, encourages the allocation of mass at highly connected nodes, while the second term $L_2$ encourages most of the mass of $\mathbf{p}$ to be contained in a clique. Moreover, $\beta$ is a hyper-parameter which is used to balance the contributions of the two terms. As mentioned, an almost identical loss function was already used in Karalias and Loukas [2020] with further details discussed in Appendix C.

Similar to many other GNN applications it is desirable to use the sparsity of the graph G and hence also $\mathbf{W}$. We note that, although a sparse $\mathbf{W}$ signifies a dense $\overline{\mathbf{W}}$, we can efficiently evaluate the second loss term as $\mathbf{p}^T \overline{\mathbf{W}}\, \mathbf{p} = (\sum_{v=1}^n \mathbf{p}_v)^2 - \mathbf{p}^T \mathbf{W}\, \mathbf{p} - \sum_{v=1}^n \mathbf{p}_v^2$.

---

**Algorithm 1** Maximum Clique Decoder

---

1: **Input:** Probability vector $\mathbf{p}$, graph $G$ with $n$ nodes, number of samplers $\kappa$, sample length $\tau$.
2: **Output:** Predicted maximum clique $\hat{C}$ of size $|\hat{C}|$.
3: Order nodes via permutation $\pi : V \to [n]$ such that $\mathbf{p}_{\pi^{-1}(1)} \geq \mathbf{p}_{\pi^{-1}(2)} \geq \cdots \geq \mathbf{p}_{\pi^{-1}(n)}$.
4: **for** $j = 1$ **to** $\kappa$ **do**
5:     $\hat{C}_j \leftarrow \{\pi^{-1}(j)\}$
6:     **for** $i = 2$ **to** $\tau - \kappa$ **do**
7:         **if** $\hat{C}_j \cup \{\pi^{-1}(j+i)\}$ forms a clique **then**
8:             $\hat{C}_j \leftarrow \hat{C}_j \cup \{\pi^{-1}(j+i)\}$
9:         **end if**
10:    **end for**
11: **end for**
12: $\hat{\Omega} \leftarrow \{\hat{C}_1, \ldots, \hat{C}_\kappa\}$
13: $\hat{C} \leftarrow \arg\max_{C \in \hat{\Omega}} |C|$

---

### 3.5 Decoder

In order to extract the maximum clique from $\mathbf{p}$, we use a greedy decoder detailed in in Algorithm 1. Our algorithm creates a set $\Omega = \{\hat{C}_j\}_{j=1}^{\kappa}$ of $\kappa$ cliques and then selects the $\hat{C}_j$ with largest cardinality. To construct each $\hat{C}_j$, we first create a sorting function $\pi : V \to \{1, \ldots, |V|\}$ which puts the nodes in descending order, so $\mathbf{p}_{\pi^{-1}(1)} \geq \ldots \geq \mathbf{p}_{\pi^{-1}(n)}$. Each $\hat{C}_j$ starts as a single node $\pi^{-1}(j)$. We then consider $\tau - j$ additional candidates, $\pi^{-1}(j+1), \ldots, \pi^{-1}(\tau)$ and accept each candidate node $v_{\text{canditate}}$ if $\hat{C}_j \cup \{v_{\text{candidate}}\}$ forms a clique. This is tested by verifying that $\mathbf{x}^\top \overline{\mathbf{W}} \mathbf{x} = 0$ for $\mathbf{x} := \sum_{v \in \hat{C}_j \cup \{v_{\text{candidate}}\}} \mathbf{1}_v$. We remark that in order for our decoder to predict the true maximum clique exactly, we must choose the parameters $\tau$ and $\kappa$ such that there exists a $j$, $1 \leq j \leq \kappa$, such that $j \leq \pi^{-1}(i) \leq \tau$ for all $v_i$ in the MC. On simple datasets, such as IMDB, it suffices to set $\kappa = 1$. More complex datasets such as Twitter require larger values of $\kappa$. Importantly, we note that the $\hat{C}_j$ can be computed in parallel, so increasing the value of $\kappa$ does not create major scalability issues. We note that setting the hyperparameters of the decoder requires some prior knowledge of the largest sizes of the maximum cliques present in the data, which we tune using the validation set.

## 4 Results

### 4.1 Baselines

We compare our hybrid scattering model against other neural network-based methods, an integer programming-based solver and a traditional heuristic. For the the neural networks, Erdös' GNN [Karalias and Loukas, 2020] and RUN-CSP [Toenshoff et al., 2021], the authors provide two implementations, one optimized towards computational speed (fast), and the other optimized for best approximation score (accurate). In order to highlight the contribution of our hybrid scattering model, we also provide results for a pure low-pass model, which only uses GCN-type filters [Kipf and Welling, 2016] followed by our decoder module. We further report results for the integer-programming method Gurobi 9.0 [Gurobi Optimization, LLC, 2022]. Finally, we compare to a traditional heuristic [Grosso et al., 2008], a local search procedure with $\eta_1$ random restarts, where an initial clique is continuously modified and possibly improved by neighbourhood search operations until some termination condition occurs. The heuristic is iterated for a maximum of $\eta_2$ iterations. We denote the method for different hyperparameter configurations as Heuristic$(\eta_1, \eta_2)$.

### 4.2 Datasets

We evaluate our model on three popular real-world graph learning datasets, namely IMDB, COLLAB [Yanardag and Vishwanathan, 2015] and TWITTER [Yan et al., 2008]. To further explore how our model performs relative to other methods on more challenging and larger graphs, we introduce two new datasets consisting of graphs generated according to the approach in Xu et al. [2005]. The first dataset (SMALL) contains graphs of slightly larger size ($\sim$180 nodes on average) compared to the

Table 1: Maximum clique test approximation score (mean ± std.) and avg. prediction time measured in seconds per graph (in brackets) on real-world datasets compared to the different baselines. In our decoder, we set $\kappa$ equal to 1, 1 and 10 for IMDB, COLLAB and TWITTER, respectively. We provide results for Gurobi 9.0 with 4 different time budgets, and for the heuristic with different configurations.

| Dataset | IMDB | COLLAB | TWITTER |
|---|---|---|---|
| **ScatteringClique** (ours) | 1.000 ± 0.000 (4e-3) | 0.999 ± 0.014 (0.029) | 0.952 ± 0.059 (0.05) |
| GCN (low-pass) | 0.956 ± 0.109 (3e-3) | 0.981 ± 0.085 (0.020) | 0.887 ± 0.111 (0.04) |
| Erdős (fast) | 1.000 ± 0.000 (0.08) | 0.982 ± 0.063 (0.10) | 0.924 ± 0.133 (0.17) |
| Erdős (accurate) | 1.000 ± 0.000 (0.10) | 0.990 ± 0.042 (0.15) | 0.942 ± 0.111 (0.42) |
| RUN-CSP (fast) | 0.823 ± 0.191 (0.11) | 0.912 ± 0.188 (0.14) | 0.909 ± 0.145 (0.21) |
| RUN-CSP (acc.) | 0.957 ± 0.089 (0.12) | 0.987 ± 0.074 (0.19) | 0.987 ± 0.063 (0.39) |
| Gurobi 9.0 (0.1s) | 1.000 ± 0.000 (1e-3) | 0.982 ± 0.101 (0.05) | 0.803 ± 0.258 (0.21) |
| Gurobi 9.0 (0.5s) | 1.000 ± 0.000 (1e-3) | 0.997 ± 0.035 (0.06) | 0.996 ± 0.019 (0.34) |
| Gurobi 9.0 (1s) | 1.000 ± 0.000 (1e-3) | 0.999 ± 0.015 (0.06) | 1.000 ± 0.000 (0.34) |
| Gurobi 9.0 (5s) | 1.000 ± 0.000 (1e-3) | 1.000 ± 0.000 (0.06) | 1.000 ± 0.000 (0.35) |
| Heuristic$(5, 10)$ | 0.912 ± 0.190 (1.0e-2) | 0.276 ± 0.222 (0.09) | 0.706 ± 0.188 (0.08) |
| Heuristic$(5, 20)$ | 0.972 ± 0.081 (1.5e-2) | 0.466 ± 0.274 (0.17) | 0.870 ± 0.124 (0.14) |
| Heuristic$(5, 30)$ | 0.998 ± 0.015 (1.8e-2) | 0.620 ± 0.299 (0.26) | 0.910 ± 0.094 (0.29) |
| Heuristic$(20, 20)$ | 0.973 ± 0.080 (5.8e-2) | 0.469 ± 0.277 (0.70) | 0.940 ± 0.098 (0.61) |
| Heuristic$(1, 100)$ | 0.919 ± 0.133 (6.0e-3) | 0.847 ± 0.257 (0.16) | 0.689 ± 0.260 (0.07) |
| Heuristic$(5, 100)$ | 0.997 ± 0.021 (2.7e-2) | 0.916 ± 0.201 (0.77) | 0.920 ± 0.084 (0.36) |

TWITTER dataset (∼132 nodes on average), which is the real-world dataset with the largest graphs. The second dataset (LARGE) contains significantly larger graphs (∼1324 nodes on average), that are far more challenging. In each of the two datasets, we have three different classes of hardness (easy, medium and hard) that are retained for different hardness parameters (0.2, 0.5 and 0.8) following the approach in Xu et al. [2005]. More details on the datasets can be found in Table 3 in Appendix B.

## 4.3 Evaluation

To evaluate the different models, we use average test approximation score (mean ± standard deviation across three runs) and the average time needed to approximate the MC across the graphs of a dataset (in brackets) measured in seconds per graph (s/G). We note that the time requirement of Gurobi 9.0 can exceed the indicated time limit as it only affects a subroutine of the method. On the real-world datasets and the proposed SMALL dataset, the approximation score of a given graph is computed by dividing the size of the clique found by the algorithm by the size of the true MC, i.e., if the MC of a graph has size 10 and the algorithm returns a clique of size 9, it would be given a score of 90%. We then average these scores across all graphs in each dataset. On the proposed LARGE dataset, the MC size is unknown because it is virtually impossible to specify. Instead, we use the predicted MC size from Gurobi 9.0 (0.1s) as a target. Notably, this allows for approximation scores larger than 100% to arise, which signifies that the model found a better (larger) clique than Gurobi 9.0 (0.1s).

## 4.4 Experiments

We first look at the three real-world datasets and provide results in Table 1. In the decoder (Section 3.5), we set $\kappa$ equal to 1, 1 and 10 for the IMDB, COLLAB and TWITTER dataset, respectively. For the IMDB dataset, our hybrid approach and most other methods (except RUN-CSP) solve the MC problem with a perfect (100%) approximation score. However, our model significantly outperforms the other neural network-based models (Erdös' GNN and RUN-CSP) with respect to computation time (although the highly optimized Gurobi solver is slightly faster than our method on this dataset due to the small graph size). On COLLAB, our model almost reaches perfect performance (99.9%) in 0.029 s/G, outperforming the other neural network based methods in both speed and approximation score. Some versions of the Gurobi solver are able to achieve 99.9% and 100% approximation scores, however, they are slower than our method on this dataset. The TWITTER dataset is the most challenging of the considered real-world datasets. Here, our method achieves the second highest accuracy among neural networks with 95.2% compared to 98.7% for RUN-CSP (accurate). However, our method is nearly seven times faster than RUN-CSP (0.05 s/G vs 0.34 s/G). Similarly, while

there are several versions of Gurobi which achieve higher accuracy, these require at least 0.34 s/G. In principle, one could utilize faster versions of Gurobi, but those methods would achieve lower accuracies. For example, Gurobi 9.0 (0.1s) requires four times the time per graph compared to our method (0.21 s/G vs 0.05 s/G) while achieving a much lower accuracy (80.3% vs 95.2%).

Figure 3 illustrates the differences between our hybrid scattering model and the low-pass GCN-based model [Kipf and Welling, 2016] and helps explain why the scattering-based model performs better across all three datasets. It shows that the probability vector $\mathbf{p}$ generated by our approach is more discriminative than the one produced by the low-pass model. When using the low-pass model, most nodes are assigned high probabilities (marked in red). The decoder cannot tell which nodes have higher priorities given such a smooth output. This leads to the decoder accepting nodes which are not part of the MC. If the new clique is not a sub-clique of the MC, this in turn can lead to the decoder rejecting nodes which *are* part of the MC. The scattering model on the other hand produces a less smooth representation. The nodes outside the MC are assigned lower probabilities so that our decoder can successfully find the MC. This is consistent with previous work [Wenkel et al., 2022] showing that graph scattering helps to alleviate oversmoothing in node classification. We emphasize that both models are trained with the same loss function and the same rule-based decoder. Therefore, the differences illustrated in Figure 3 are the direct result of using a hybrid scattering model. More comparison between the scattering model and low-pass model can be found in Appendix E.

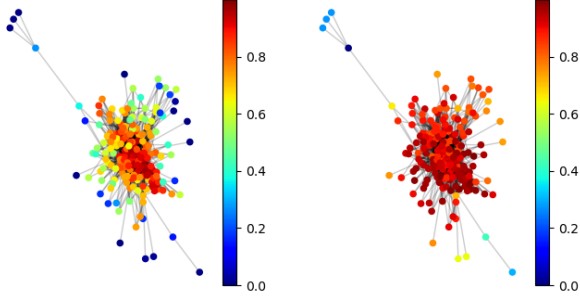

Figure 3: Comparison of the output probability vector $\mathbf{p}$ for our model (left) and the low-pass model (right) on a graph taken from TWITTER with ground truth MC of size 10. Our approach followed by our decoder yields the correct clique, while the low-pass model concludes with a clique of size 8.

To further explore how our model performs on *harder* graphs, we now look at the newly proposed datasets generated according to the methodology of Xu et al. [2005]. Each dataset contains three subclasses (easy, medium and hard) and our decoder (Section 3.5) uses different parameters $\kappa$ (1, 1 and 10), respectively. Here, we present the results of the most challenging LARGE dataset, shown in Table 2, while the discussion of the results on the SMALL dataset can be found in Appendix D.

Our hybrid scattering model outperforms the other neural network approaches on all three subclasses (easy, medium and hard) in terms of approximation score and inference time. The gap in approximation score is larger on medium and hard samples, highlighting the utility of our model for more complex graphs. At the same time, our model is extremely efficient, exhibiting much lower inference times than the other neural networks (with the exception of GCN, which is a modified version of our hybrid model with band-pass filters removed), e.g., inferring the MC on LARGE-hard 12 times faster than RUN-CSP (accurate), the closest competing neural network. For sufficiently large time budgets, Gurobi and the heuristic [Xu et al., 2005] can achieve higher accuracies, however at a significantly higher time cost. For example, on LARGE-hard, only Gurobi 9.0 (50.0s) and $\mathrm{Heuristic}(5, 100)$ are better, however taking on average 131 and 49 times longer to infer the MC, respectively.

Overall, across all considered datasets, we remark that our hybrid model provides considerable speedup compared to both the other neural networks and to Gurobi for large graphs. We also note that our model takes fewer parameters than the Erdös' GNN [Karalias and Loukas, 2020]. They use a multi-head GNN (8 heads, with 64 hidden units each) for each layer, whereas the MLPs used in our method have $d_h = 8$ hidden units on all three datasets, without using any multi-head attention mechanism. This in turn suggests that the use of both GCN filters *and* scattering filters combined yields powerful feature extractors that help learn informative node representations with relatively few

Table 2: MC test approx. score (mean $\pm$ std.) and avg. prediction time measured in seconds per graph (in brackets) on LARGE dataset for our model compared to the different baselines. In our decoder, we set $\kappa$ equal to 1, 1 and 10 for the easy, medium and hard graphs, respectively. As ground truth sizes cannot be estimated for such large graphs, we compare to the cliques found by Gurobi 9.0 (0.1s). As explained in Sec. 4.3, the inference time of Gurobi can exceed the indicated time budget.

| Dataset | LARGE-easy | LARGE-medium | LARGE-hard |
|---|---|---|---|
| **ScatteringClique** (ours) | $1.002 \pm 0.053$ (0.25) | $1.045 \pm 0.171$ (0.18) | $1.102 \pm 0.225$ (0.46) |
| GCN (low-pass) | $0.997 \pm 0.088$ (0.23) | $0.925 \pm 0.181$ (0.16) | $0.829 \pm 0.302$ (0.40) |
| Erdős (fast) | $0.969 \pm 0.103$ (1.08) | $0.924 \pm 0.210$ (1.61) | $0.895 \pm 0.246$ (1.96) |
| Erdős (accurate) | $0.994 \pm 0.067$ (4.74) | $0.943 \pm 0.113$ (5.62) | $0.975 \pm 0.086$ (5.50) |
| RUN-CSP (fast) | $0.925 \pm 0.082$ (1.65) | $0.895 \pm 0.173$ (2.53) | $0.985 \pm 0.134$ (1.44) |
| RUN-CSP (accurate) | $0.985 \pm 0.072$ (4.72) | $0.913 \pm 0.169$ (4.69) | $1.034 \pm 0.159$ (5.58) |
| Gurobi 9.0 (0.1s) | $1.000 \pm 0.000$ (10.5) | $1.000 \pm 0.000$ (10.6) | $1.000 \pm 0.000$ (9.46) |
| Gurobi 9.0 (0.5s) | $1.000 \pm 0.000$ (12.1) | $1.000 \pm 0.000$ (11.0) | $1.000 \pm 0.000$ (9.97) |
| Gurobi 9.0 (1.0s) | $1.000 \pm 0.000$ (12.7) | $1.000 \pm 0.000$ (11.1) | $1.000 \pm 0.000$ (10.1) |
| Gurobi 9.0 (5.0s) | $1.000 \pm 0.000$ (16.4) | $1.000 \pm 0.000$ (14.7) | $1.000 \pm 0.000$ (14.2) |
| Gurobi 9.0 (20.0s) | $1.005 \pm 0.070$ (30.0) | $1.094 \pm 0.290$ (29.4) | $1.016 \pm 0.122$ (29.0) |
| Gurobi 9.0 (50.0s) | $1.009 \pm 0.082$ (65.9) | $1.276 \pm 0.431$ (55.4) | $1.335 \pm 0.459$ (60.3) |
| Heuristic$(5, 10)$ | $0.326 \pm 0.039$ (1.6) | $0.331 \pm 0.050$ (2.1) | $0.311 \pm 0.072$ (2.2) |
| Heuristic$(5, 20)$ | $0.619 \pm 0.008$ (3.2) | $0.617 \pm 0.097$ (4.5) | $0.575 \pm 0.130$ (4.6) |
| Heuristic$(5, 30)$ | $0.895 \pm 0.008$ (4.9) | $0.880 \pm 0.121$ (6.7) | $0.813 \pm 0.185$ (7.1) |
| Heuristic$(20, 20)$ | $0.623 \pm 0.075$ (12.9) | $0.630 \pm 0.095$ (17.8) | $0.593 \pm 0.136$ (18.5) |
| Heuristic$(1, 100)$ | $0.975 \pm 0.050$ (2.7) | $0.972 \pm 0.259$ (3.7) | $0.948 \pm 0.399$ (4.6) |
| Heuristic$(5, 100)$ | $0.999 \pm 0.003$ (13.8) | $1.149 \pm 0.351$ (18.8) | $1.296 \pm 0.483$ (22.9) |

parameters. In fact, the Erdös' GNN model contains 1,880,709 parameters for each dataset, while we only use 1,297 parameters for Twitter, 705 for IMDB and 1,001 for COLLAB, respectively. This corresponds to 0.07%, 0.04% and 0.05% of the parameter counts of Erdös' GNN, while achieving better performance in terms of both inference time and accuracy. Further, since the $K$-layer GCN-type model has time complexity $O(K|E|d_h + Knd_h^2)$ [Wu et al., 2020], it is important to reduce $d_h$. We use $d_h = 8$ (and no multi-head attention), resulting in low time complexity and thus faster inference.

## 5 Conclusion

In this paper, we show the expressive power of GNNs can be critical for solving graph CO problems. Our results suggest that low-pass models that generally enforce smoothness over graph neighborhoods due to oversmoothing, make the nodes indistinguishable, which leads to unfavorable node representations for the MC problem. Inspired by the geometric scattering transform, we propose an efficient scattering-based GNN to overcome the oversmoothing problem. We use a two-term loss function (previously derived by a different method in Karalias and Loukas [2020]), which encourages our network to find a set of nodes that are both highly connected and contained within a clique, which in conjunction mimic the otherwise hard to optimize problem of finding maximum cliques. Our model produces efficient node representations that enable us to approximate the maximum clique at a fast speed using a constrain-preserving greedy decoder. Overall, our performance is competitive with other neural network-based methods on common benchmark datasets, while largely reducing the number of parameters and computation time. When run on larger and more complex graphs, our hybrid scattering model outperforms other neural network approaches.

## 6 Acknowledgements

We would like to thank Yiwei Bai for discussions. This work was partially funded by Fin-ML CREATE graduate studies scholarship for PhD [Frederik Wenkel]; IVADO (Institut de valorisation des données) grant PRF-2019-3583139727, FRQNT (Fonds de recherche du Québec - Nature et technologies) grant 299376, Canada CIFAR AI Chair; and NIH (National Institutes of Health) grant NIGMS-R01GM135929 [Guy Wolf]. The content provided here is solely the responsibility of the authors and does not necessarily represent the official views of the funding agencies.

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
