# OpenReview forum: "Can Hybrid Geometric Scattering Networks Help Solve the Maximum Clique Problem?"
_NeurIPS.cc/2022/Conference — NeurIPS 2022 Accept_

### Official Review · Reviewer_fuxA · 2022-06-29

**Rating:** 3
**Confidence:** 4
**Soundness:** 2 fair
**Presentation:** 2 fair
**Contribution:** 1 poor

**Summary:**

The paper proposes a novel model that is trained without supervision to solve the maximum clique problem. Furthermore, a greedy decoding scheme is proposed to enable the discretization of the continuous output from the neural network. The model architecture relies on the scattering transform which has been proposed to avoid oversmoothing in GNNs. The proposed model achieves competitive results on common experimental benchmarks and on synthetically generated instances of varying difficulty.

**Questions:**

1) Did the authors use the RB model to generate the instances of table 3? Did the authors code it up from scratch? I could not find the implementation in the code provided. More details about the exact setup would be appreciated.
2) Apart from over-smoothing, does the scattering architecture offer a particular benefit for the maximum clique problem? It is not clear to me whether the choice of this architecture relates to this particular problem or not.


**Limitations:**

The authors have not discussed the limitations or the societal impact (arguably not applicable here) of their work.

**Strengths And Weaknesses:**

## Strengths
1) The approach is simple and well explained in the paper.
2)  The authors evaluate on an experimental setup from the literature, making comparisons with related work easier.
3) Results appear to be competitive across multiple datasets.

## Weaknesses and criticism
1) Parts of this paper's pipeline are not properly attributed to previous work. For example, the loss function includes two terms:
the expected weight of the edges in the selected set, and the expected weight of the edges on the complement graph.
This is essentially the loss function used in the paper by Karalias and Loukas (henceforth KL for brevity) that the authors cite throughout the paper, but not specifically for the loss. In fact, line 217 in the conclusion claims "We further construct a novel two-term loss function...". I think it is fair to say that this construction is not novel.
The loss function may appear to be different from the one in KL but is actually extremely similar, minus a few additional steps that are required in KL to secure the theoretical guarantee. To understand the similarity, it suffices to check the proof of corollary 1 in the appendix of that paper. The probability of constraint violation amounts to the expected weight on the complement graph (denoted by  the expectation of $\bar{w}(S) $).
However, that paper rewrites the term as a function of the graph itself and avoids the computation in the complement. This is the main reason behind the apparent differences between the two loss functions. Using both the original graph and its complement has the inevitable drawback of not exploiting the sparsity of the graph since at least one of the two will inevitably be dense. Ultimately, the proposed loss is a slightly simplified rewriting of the KL loss.

2) The authors emphasize the over-smoothing problem, but this can be overcome with various architectures and/or skip connections. Indeed, both the Erdos paper and RUN-CSP do not just use plain GCN architectures and manage to do fairly well. I am not entirely convinced that the proposed model offers a substantial benefit.
Perhaps some ablations that demonstrate the over-smoothing problem on existing implementations of combinatorial ML papers from the literature would have helped.

3) Minor issue: the authors mention the "maximal clique" problem although I think they mean to say "maximum". Given that the authors mention the hardness of the problem and approximation ratios, I assume that maximum is the term that applies (maximal cliques can be found easily).

4) Experimentally, it would be good to see how the other techniques (e.g., Erdos or RUN-CSP) perform on Xu instances (table 3).  A greedy heuristic would also be nice as well.

5) Apart from the different model, this approach does not significantly differ from works like the KL paper or even RUN-CSP, with the added limitation that it only addresses the maximum clique problem.

6) Scalability is not addressed in the paper. Does the proposed model scale well on larger graphs? The instances that this was tested on have at most a few hundred nodes. Given that the paper places heavy emphasis on the model, more thorough experimental demonstrations are required in that regard as well.

---

> ### Author Response · Authors · 2022-08-02
> **response to Reviewer fuxA**
>
> Please check "Response to all reviewers" first:
>
> **Weaknesses and criticism**
> 1. […the two loss functions. Using both the original graph and its complement has the inevitable drawback of not exploiting the sparsity of the graph…]: First, this loss term is calculated in the training time, so it will not affect the model’s inference speed. Second, the loss term $L_2$ can be written as $ p^T \overline{W} p = (\sum_{v=1}^n p_v)^2 - p^T W p - \sum_{v=1}^n p_v^2$, so that no dense graph computation is required.
> […slightly simplified rewriting of the KL loss…]: The initial KL loss contains two parts: the first term is conditioned on the clique, while our loss is not. The second term in KL paper does not depend on the edge matrix, while the second term in our loss term is the complement of the edge matrix, which is based on the edge matrix. So the loss term is NOT just a simplified rewriting.
> Our new loss term and the scattering structure enable our model to successfully outperform KL paper in both time and accuracy, using around 1/1000 of the parameters that KL takes. Considering the number of parameters our model takes, we now change the term ‘novel’ to ‘efficient’.
> 2. […I am not entirely convinced that the proposed model offers a substantial benefit…]: Our scattering model is very parameter-efficient and uses around 0.1% parameters of the previous baseline (KL paper). Note that though there are various architectures, like adding skip connections may also works, but introducing skip-connections has no evidence to reduce the model’s complexity, in fact, it may hurt the model’s inference time. (because usually, skip-connection structures are very deep and wide, which increases the time complexity.) Our new draft explains why the scattering model offers a substantial benefit. Because the running time of a GCN-type model is quadratic to the width of the model’s hidden space, that is to say, increasing the width of GNN or using a multi-head mechanism will result in larger time complexity. Previous KL (erdos GNN) paper uses 8 head GIN (Graph Isomorphism Network) with 64 hidden units and the model consists of over 1.8 million parameters. In contrast, our scattering does not use a multi-head mechanism and uses only 8 hidden units and takes around 1,000 parameters.
> 3. […I assume that maximum is the term that applies…] We change maximal to maximum.
> 4. [...other techniques … greedy heuristic would …]: We add new results and discussion on the difference between our method and the heuristic.
> 5. [...significantly differ from works like the KL paper or even RUN-CSP…]: Our model takes only 1/1000 parameters of the previous baseline model and achieves better results and runs faster, we remark our results as highly non-trivial, and our structure is significantly different from the previous one. In this paper, we propose this highly efficient method and run experiments on both natural datasets and tasks with different hardness. In practice, any graph combinatorial problem that can be written as in True versus False fashion, such as mac cut, vertex cover, etc, can also use our baseline. We leave these tasks for future work.
> 6. [... on larger graphs….]: we add experiments on large graphs. Note on large graphs, we are not able to get the ground truth solutions, we use Gurobi (0.1s) as baseline.
>
> **Questions:**
>
> 7. [...More details about the exact setup …]: we add the code in the github repo.
> 8. [...particular benefit …]: our scattering overcomes the over-smoothness problem, also our model is very parameter efficient, which takes only ~0.1 % of previous baseline, since we (or most heuristics) aim to quickly obtain a good approximation of the MC. An efficient structure with expressive power is very important. In conclusion, there are following three benefits: 1.Overcome the oversmoothing 2.Faster running time with higher accuracy 3.Reduce the parameter counts of the previous model by over 99.9%
>
> **Limitations:**
>
> 9.  [... limitations or the societal impact…] The limitation of our approaches: our GNN model focuses on maximum clique. We don’t see any direct negative societal impact here. When generalizing this method to other graph structures such as social networks or biological data, we should protect personal information/privacy in that dataset.

---

> > ### Comment · Reviewer_fuxA · 2022-08-02
> > **response to point 1**
> >
> > Regarding point 1:
> >
> > You claim that the loss is not a simplified rewriting, so I need to understand which of the following you disagree with:
> >
> > 1) Your loss is given by equation 11 and it's $L(\mathbf{p}) = -\mathbf{p}^\top \mathbf{W}\mathbf{p}  + \beta \mathbf{p}^\top \bar{\mathbf{W}}\mathbf{p} $.
> > 2) The KL probabilistic penalty loss is described by equation 3 in that paper:
> > $  \mathbb{E} [f(S)] + \beta P(S \notin \Omega) $.
> > Let's  follow their derivation on appendix D.3.1.
> > 3) $  \mathbb{E} [f(S)] =   \gamma - \mathbb{E} [w(S)] = \gamma -  \sum_{i,j} w_{ij}p_ip_j =\gamma -  \mathbf{p}^\top \mathbf{W}\mathbf{p} $
> > 4) From that same section we have for the constraint  $P(S \notin \Omega) \leq     \mathbb{E} [\bar{w}(S)] $. This is the expected weight on the *complement graph*, i.e., on the complement of the edge matrix as you write.
> > 5) $ \mathbb{E} [\bar{w}(S)] =  \mathbf{p}^\top \bar{\mathbf{W}}\mathbf{p} $ because this is the expected weight of $S$ on the complement graph. So this quantity can be used to bound $P(S \notin \Omega)$. The authors continue a few steps and do some additional manipulations on this expression to arrive at their final loss.
> > 6) Putting it all together: $  \mathbb{E} [f(S)] + \beta P(S \notin \Omega)  \leq \gamma -  \mathbf{p}^\top \mathbf{W}\mathbf{p} + \beta \mathbf{p}^\top \bar{\mathbf{W}}\mathbf{p}. $
> >
> > Point 6 has the Erdos loss while point 1 has the loss proposed in your paper. I believe this illustrates how your loss is essentially a simplification that gets rid of the displacement term $\gamma$  (it doesn't matter for optimization anyway) and does not do the extra steps the authors of that paper do in the appendix to arrive at the particular final version of the loss in their Corollary 1.  Those steps are just rewriting the expression so qualitatively your loss is a simplified rewriting. Maybe you disagree with my characterization but I hope it's clear from points 1 and 6 that the losses are almost the same which is why I challenged the claim of "novelty" for the loss function.
> >
> > Does this make sense?

---

> > > ### Author Response · Authors · 2022-08-03
> > > **response to the loss:**
> > >
> > > Dear reviewer, in KL paper, the authors are using additional Markov’s inequality to bound the loss.
> > >
> > > see equation 12 in KL's supplementary material, where they write:
> > >
> > > $P(S \not\in \Omega) \leq \frac{1}{2} \sum_{v_i \neq v_j} p_i p_j$ - $\mathbb{E} [w(S)]$.
> > >
> > > and here we are using:
> > >
> > >  $P(S \not\in \Omega) = p^T \overline{W} p$.
> > >
> > > They are not the same term(s).

---

> > > > ### Comment · Reviewer_fuxA · 2022-08-03
> > > > **loss**
> > > >
> > > > 1) I used Markov's as well. When I say $P(S \notin \Omega) \leq \mathbb{E}[\bar{w}(S)]$, that's a consequence of Markov's inequality....I'm just going a few lines up from the line you quoted in equation 12.  The points I wrote to you are not my derivation. I'm copying them from the paper but I stop before the rewriting they do to explain to you that what you have is almost the same.
> > > > Not *exactly* the same because you don't use explicitly the probabilistic method so you have no need to invoke Markov's inequality etc. and you are carrying an extra factor of 2 compared to their derivation.
> > > >
> > > > 2) Let me provide the precise derivation.
> > > > You say that your term is $P(S \notin \Omega) = \mathbf{p}^\top \bar{\mathbf{Wp}}$.  We know that  $\bar{\mathbf{W}} = \mathbf{1} - (\mathbf{I} + \mathbf{W})$, where $\mathbf{1}$ is an $n \times n$ all ones matrix.
> > > > So we have
> > > > \begin{align*}
> > > > \mathbf{p}^\top \bar{\mathbf{Wp}} &= \mathbf{p}^\top (\mathbf{1} - \mathbf{I})\mathbf{p} - \mathbf{p}^\top \mathbf{Wp} \newline
> > > > &= \sum_{v_i \neq v_j} p_ip_j  - 2\sum_{(v_i,v_j) \in E} p_i p_j w_{ij} \newline
> > > > &= \sum_{v_i \neq v_j} p_ip_j - 2\mathbb{E}[w(S)]  \tag{$\mathbb{E}[w(S)] = \sum_{(v_i,v_j) \in E} p_i p_j w_{ij}$, equation 11 in KL} \newline
> > > > &=  2( \frac{1}{2}\sum_{v_i \neq v_j} p_ip_j - \mathbb{E}[w(S)]). \tag{this is the KL expression scaled by 2}
> > > > \end{align*}
> > > >
> > > > It is clear that this is the expression in equation 12 of the KL paper scaled by a factor of 2. The one that you wrote in your comment.
> > > > I hope we can agree that one is essentially a scaled up version of the other.
> > > >
> > > >
> > > > P.S.: In my previous comment, to make the similarity more apparent, I suppressed the factor of 2, i.e., I took $\mathbb{E}[w(S)] = \mathbf{p}^\top \mathbf{Wp}$, when it's technically $\mathbb{E}[w(S)] = \mathbf{p}^\top \mathbf{Wp}/2$ in the KL paper. Maybe that was the source of confusion?

---

> > > > > ### Author Response · Authors · 2022-08-03
> > > > > **loss**
> > > > >
> > > > > We acknowledge the reviewer to clarify the source of confusion, yes we now agree that we are using a scaled up version.
> > > > > In the updated manuscript, as said in the 1st rebuttal, we already removed the word 'novel'.
> > > > >
> > > > > Since this paper's main idea is to introduce the scattering model and focus on oversmoothing, parameters-efficiency and approximation performance. Exploring how "scale up loss V.S. non-scale up loss" will affect the performance seems a little bit off the track. We suspect that using the KL loss, which is $\frac{1}{2}$ of our loss, will make the approximation ratio change very slightly but not affect the running speed and the number of parameters. We are happy to add discussion and clarification of the loss terms to our manuscript. This discussion and clarification can also help readers to understand the model better.

---

> > ### Comment · Reviewer_fuxA · 2022-08-03
> > **Overall**
> >
> > - My point regarding the loss being extremely similar to related work stands. The authors have confirmed that they will update the relevant parts of the submission accordingly.
> > - The authors have addressed some of my concerns regarding the model. Indeed it appears that the model is a lightweight solution that performs well compared to previous work and there lies the main contribution of the paper.  On the other hand, there exist published versions of scattering models (like the scattering gcn published at neurips a few years ago that also aims to address oversmoothing), and in terms of approximation ratios, the model achieves mostly marginal improvements, with the exception of the Xu-type instances where there is a more noticeable benefit.
> >
> > Given the above and that the model is only shown to work on one combinatorial problem, I do not think there's enough of a contribution to the field of Combinatorial Optimization + ML to warrant acceptance so I maintain my score.

---

> > > ### Author Response · Authors · 2022-08-04
> > > **response to 'overall'**
> > >
> > > We are glad that some of your concerns have been addressed.
> > > And we are happy that the reviewer agrees with us on the following points:
> > > 1. Our model is lightweight (~0.1 % parameters count) and performs well compared to previous work.
> > > 2. We get a more noticeable benefit on the hardness dataset.
> > > 3. Our structure addresses the oversmoothing.
> > > 4. The loss is similar
> > >
> > > **Regarding the loss:**
> > >
> > > this paper discusses how oversmoothing will affect the performance and how different structures (number of parameters) will affect the inference speed. We do not discuss different types of loss in this paper. __Furthermore, we are glad that the reviewer describes our loss as 'extremely similar to related work.' This further strengthens the claim that our expressive network structure brings substantial improvement to the MC task. (instead of the loss function)__
> > >
> > > **Regarding the performance:**
> > >
> > >  the reviewer says that our model ```'achieves mostly marginal improvements... but also a noticeable benefit on Xu-type instances.'``` As shown in our paper, the approximation ratio of baseline model on the first three datasets is already very high (>90%), which leaves little room for improvement.  __To show that our model has a substantial improvement, we introduce Xu-type instances, including experiments with large graphs, see the supplementary material.__ We further explain why our method runs faster than the traditional heuristic with the restart strategy.
> > >
> > > **Regarding scattering:**
> > >
> > > First, our scattering structure is different from scattering GCN. We are introducing new modules, such as read-out and a new type of attention mechanism, etc.
> > > Second, one of the primary goals of designing a scattering model is to __'show the expressive power of GNNs can be critical for solving graph CO problems'__, as stated in the introduction and the first line of our conclusion. And we think the reviewer also agrees with that.
> > >
> > > **Regarding 'shown to work on one combinatorial problem:'**
> > >
> > > Due to the page limit, we are discussing the maximum clique in this paper. Our works show that an efficient structure with expressive power (no oversmoothing) is important. We think this claim holds for any bi-variable graph CO problems because we need to consider that time cost and separate 1 and 0 (True and False). We leave the extension for future work.

---

### Official Review · Reviewer_7b6z · 2022-07-08

**Rating:** 6
**Confidence:** 4
**Soundness:** 2 fair
**Presentation:** 3 good
**Contribution:** 2 fair

**Summary:**

The maximum clique problem is a classic NP-hard combinatorial optimization problem with numerous applications. Because of its difficult, heuristics that find good solutions fast are desirable, and there has been a few works that have explored machine learning for designing such algorithms. In this paper, the authors propose a novel approach based on a GNN that is trained by supervised learning to predict the probability of a node to belong to a maximal clique, followed by a greedy algorithm (a decoder) which constructs as large a clique as possible from the probabilities. In departure from previous work, they propose that the model use a geometric scattering transform, which reduces neighbor smoothing. They show improvements on empirical datasets against alternative neural network approaches, and Gurobi with a time limit.

**Questions:**

- I don't really understand why the decoder has a Tau parameter. Why not set Tau=infinity? Isn't the goal to have a large a clique as possible?
- I don't understand why the time limits given to Gurobi are not respected. Ex. in table 1, how come Gurobi with a 0.1s time limit takes 0.21s?
- Why are the competing neural network approaches are missing from Table 3?
- A more theoretical question regards non-uniqueness of solutions: maximum clique problems, especially unweighted, can often have many optimal solutions. When this is the case, won't the unsupervised loss try to steer the GNN towards an average of the solutions, leading to something which is not a maximal clique?
- I don't understand why approximation ratio is chosen as a ranking criterion in Table 1, but time is chosen as a ranking criterion in Table 3. I feel there is some cherry-picking here to make the method look best.


**Limitations:**

- I see that the method is often not competitive with Gurobi with a time limit, which makes me believe that the results would be even less competitive against human-designed heuristics for the problem. Right now, I don't think the authors really address this, which I think they should. At minimum, I think the results are too mixed to assert that the method is "competitive with commercial solvers in time and accuracy".

**Strengths And Weaknesses:**

Strengths

- The method looks faster than competing machine learning methods, at comparable performance (although this needs to be nuanced by some issues I have about the experimental results, which I detail in the weaknesses section.)
- The geometric scattering transform idea makes sense, although I am not sure whether there is something specific here about the MC problem: I would feel like the explanation given in this work would apply to many other GNN-based heuristics for graph-based combinatorial problems (e.g. vertex cover, independent set, etc.) But this is a good thing I suppose.
- The fact that the proposed training loss is unsupervised, yet differentiable, is an advantage, although I wonder what would happen if the GNN was trained in a supervised fashion to approximate pre-computed maximal cliques.

Weaknesses

- I have several issues with the experimental results. First, I find the benchmarks a little too easy: since the approximation ratio is your criterion, it is difficult to disentangle the different methods if the ratios are so high. This is particularly problematic for the IMDB dataset.
- Second, I don't really find the bolding very fair. Why highlight the two best methods in Table 1? I find it difficult to not see it as a way to hide the fact that your method is not the best on the hardest dataset (Twitter).
- I don't understand why no "normal" maximal clique heuristics are included - for example see Grosso et al. (2008). Instead you only have results against Gurobi with a time-limit, which is not really designed for your objective at hand (finding as good solutions as possible, fast). Although I can understand that it might be difficult to improve over state-of-the-art human-designed heuristics, not including them makes it difficult to assess problem difficulty.


Grosso, A., Locatelli, M. and Pullan, W., 2008. Simple ingredients leading to very efficient heuristics for the maximum clique problem. Journal of Heuristics, 14(6), pp.587-612.

---

> ### Author Response · Authors · 2022-08-02
> **Response to Reviewer 7b6z**
>
> Please check "Response to all reviewers" first:
>
> **Weaknesses:**
> 1. [...disentangle the different methods…]: We add more experiments as well as experiments on large graphs. Compared with other models, our scattering GNN can achieve a good solution quickly. The gap between running time becomes more pronounced when it comes to large graphs and graphs with medium and hard hardness. As shown in the data statistics in the supplementary material, the IMDB dataset may be too small and too easy (since every model achieves high accuracy). Except for IMDB, the scattering model always provides a good approximation within a shorter time. We also add discussion between our scattering model and traditional heuristic.
> 2. [...I find it difficult to not see it as a way to hide the fact that your method is not the best on the hardest dataset (Twitter)....]: We remove all bolding. This paper aims to obtain a fast approximation of MC, which requires us to consider running time. Note that on the Twitter dataset, RUN-CSP gets high accuracy, but the model’s running time (0.39) is larger than GUROBI (0.34) and GUROBI achieves better performance. Since the goal is to get a good solution quickly, RUN-CSP is less competitive in this case. RUN-CSP gives us the highest accuracy among all neural baselines. However, due to the RUN-CSP model’s complexity, it takes longer time than the existing solver GUROBI. This INSPIRED that we HAVE TO consider the complexity when building these neural-based heuristics. In the updated manuscript, we discuss the model’s complexity. Note that the scattering GNN model takes only ~0.1% of parameter counts of the previous baseline. Our efficient structure is the key to guaranteeing fast approximation time.
> 3. [..."normal" maximal clique heuristics are included - for example see Grosso et al. (2008)...]: We add a ‘normal’ heuristics baseline in [Grosso 2008]. We further discuss the difference between traditional local search method and our model. See the updated manuscript.
>
> **Questions:**
>
> 4. [...Tau parameter. Why not set Tau=infinity? …]: First, the clique number of such graphs is usually very close to 2 log2(n), where n is the number of nodes  [Karp 1976]. So there is no need to set \tau = infinity. Second, \tau controls the running steps of the decoder, setting \tau to a threshold saves running time. Note \tau is also used in local search heuristics, for example [Grosso 2008], where they use the parameter ‘max selection’ to control the length of iterations, which can be regarded as the parameter that controls the time and accuracy trade-off.  Reference:
> Karp, Richard M. (1976), "Probabilistic analysis of some combinatorial search problems", in Traub, J. F. (ed.), Algorithms and Complexity: New Directions and Recent Results, New York: Academic Press, pp. 1–19.
> 5. [... Gurobi are not respected…]: Note that optimization may not stop immediately upon hitting the time limit. It will stop after performing the required additional computations of the attributes associated with the terminated optimization. see
> https://www.gurobi.com/documentation/9.5/refman/timelimit.html
> 6. [ …competing neural network approaches…]: We add these results.
> 7. [A more theoretical question regards non-uniqueness ...],
> It is possible that the GNN may converge towards an average of the solutions, that is actually why we highlight the oversmoothing problem in our paper. In practice, we notice that even in the hard cases in our paper, our GNN can give you  ~ 85% of the MC size in a short time, however, pushing this accuracy higher is very difficult, even for traditional heuristics (because it’s NP-hard). One possible strategy is to design a new model that has more expressive power with non-smooth output and does not have an ‘average solution’. Another strategy is to design a term that discourages ‘average solution’ and add it to the loss function, however, for the first strategy, we need to consider the model’s complexity and for the second one, designing effective loss requires good intuition.
> 8. [...a some cherry-picking here …]: We are not choosing different criterions, the \bold notation may be confusing so we remove it. The purpose of all tables are the same: that is to show that the scattering model gets a good approximation at a very fast speed. We add other baseline models as well.
>
> **Limitations:**
>
> 9. [...often not competitive with Gurobi with…]: The key of our method is to approximate the MC size quickly, and the gap between the approximation ratio becomes more significant on larger graphs. We add comparison with human-designed heuristics. We also discuss the difference in our new manuscript. We also upload the heuristics code in the github repo. Note that all the implantation in this paper is based on python.In this paper, we propose an efficient GNN structure to fast approximate the max clique problem, we agree that the claim may be too strong, we remove this assert in the updated manuscript.

---

### Official Review · Reviewer_aPGv · 2022-07-11

**Rating:** 6
**Confidence:** 3
**Soundness:** 3 good
**Presentation:** 3 good
**Contribution:** 3 good

**Summary:**

The paper proposes to tackle the maximal clique problem in graphs with a hybrid method that relies on a scattering step as well as a rule-based decoder step to extract the predicted maximal clique. Overall, the paper is novel, clearly written and self-contained.

**Questions:**

- The approximation score seems to only measure the "size" of the MC, but not if it's correct or not? (like an overlap to the ground truth?)
please clarify why you don't measure overlap (like Jaccard) to the ground truth.

**Limitations:**

-

**Strengths And Weaknesses:**

Strengths:

- Relevant contribution to address oversmoothing of GNNs via scattering approach for MC retrieval
- Clear paper structure, the reader is taken by the hand
- The method outperforms the baselines in several datasets

Weaknesses:

- Section 1 misses some motivation: why should the general ML audience care about MC? Why is it interesting?
- Notation is a bit sloppy sometimes: Concat operation in Eq 7 not introduced, attention scores a become alpha l.109-112. element-wise operation in Eq not defined, shape dims of H_cat not defined, Objective L*(C) is a bit poorly formalized (make explicit: max_{terms to maximize over}, p \geq 0 in l.136 seems hand wavy,
- Proof of Lemma 1 is a bit quick/short: where do u_0, v_0 come from? Maybe I just missed something, but this point needs more clarification as it's not obvious to me.

- Result section could be structured more: Baselines, Evaluation metric, Results
- The approximation score seems to only measure the "size" of the MC, but not if it's correct or not? (like an overlap to the ground truth?)

Further details:
- l.124: we also want..

---

> ### Author Response · Authors · 2022-08-02
> **response to Reviewer aPGv**
>
> Please check "Response to all reviewers" first:
>
> **Weaknesses:**
> 1. [... general ML audience care about MC? Why is it interesting? …]:  we add discussion about the importance of MC problems.
> 2. [Notation is a bit sloppy sometimes: ] we revise the manuscript according to your suggestions.
> 3. [where do u_0, v_0 come from?]: u_0, v_0 come from the support of p.
> 4. [.... structured more: Baselines, Evaluation metric, Results …] We update the manuscript
> 5. [... only measure the "size" of the MC…]:  We use the ‘size’ of MC for two reasons. First, we want to be consistent with the previous papers’ evaluation metrics, (erdos gnn, RUN-CSP). Second, when the size grows larger, getting the ground truth is unrealistic because the problem is NP-hard. Actually, in our updated manuscript, we discuss the large graph cases (see the supplement material), where the ground truth is very hard to obtain. Especially when the graph size is larger than 1000, we notice that some cases do not finish in 24 hours. In this case, we can’t use ‘overlap’ as evaluation as we do not have the ground truth (we are comparing two sub-optimal results). Also, the difficulty of getting the right solutions is one of the reasons why we use unsupervised learning instead of supervised learning, as discussed in the introduction section.
>
> **Further details:**
> 6. [l.124: we also want..] fixed
>
> **Questions:**
> 7. [...only measure the "size" of the MC, but not if it's correct or not? …]: 1. We follow previous papers’ evaluation metric 2. It is very expensive to get the ground truth, (note that to make sure we get  the ‘maximum’ clique, we need to verify all possible cliques), refer to the discussion before.

---

### Author Response · Authors · 2022-08-02
**Response to all reviewers**

We thank the reviewers for their constructive comments. We notice that reviewers are concerned about the following two questions:
1. What does our scattering model differ from previous baseline and what can we benefit from the scattering model:
Answer: Our scattering model helps overcome the over-smoothing problem. It also outperforms the previous baseline model (erdos gnn) in both time and accuracy. Furthermore, the scattering model only takes ~ 0.07% parameters of the previous baseline. We remark such a reduction of parameters is a significant breakthrough.  In other words, we can reduce the parameter counts of the previous model by over 99.93% and get better performance. We think our result is highly non-trivial. Efficient GNN structure is critical for solving graph combinatorial problems because we must consider the model’s complexity. In the updated manuscript, we add a comparison between our model's complexity and the previous baseline (Erdos gnn) and explain why our model is faster.
2. Lack of enough evidence, including does not contain a heuristic:
Answer: We add more experimental results, including heuristic and results on large graphs. Our evidence indicates that scattering GNN can find a good approximation of MC at a very fast speed. Note that when the graph size grows larger, getting the ground truth is unrealistic. ( because time complexity grows exponentially)  For large graphs, we use the GUROBI solution as the baseline. We also discuss the difference between our methods and the traditional heuristic in the updated manuscript.

---

### Public Comment · ~Liangzu_Peng2 · 2022-12-01
**Related Work on Solving Maximum Clique Problems**

It is interesting to know a line of research on solving maximum cliques via deep networks.

It is important though to see a different line of research, which finds maximum cliques via optimization-based methods.

One example is https://arxiv.org/abs/1302.6256

This algorithm can solve the maximum clique problem on graphs of more than millions of nodes and edges in a few seconds (Table 1).

It seems that methods based on deep networks can solve the problem with hundreds of nodes to a certain accuracy. (Let me know if I am wrong.)

Therefore it seems essential to compare the two lines of research.

Best,
Liangzu Peng

---

### Meta-Review · Area_Chair_ipvf · 2022-08-28

**Recommendation:** Accept
**Confidence:** Certain

**Metareview:**

All reviewers agree that the proposed approach to use the geometric scattering transform is simple and effective both computationally and in terms of the ability of the method to identify larger cliques for the max-clique problem (except perhaps for one reviewer on the last point).

The work would have more impact if it could be demonstrated that using the geometric scattering transform yields improvement for other combinatorial optimization problems on graphs, or if it could outperform classical heuristics even if they are run for longer time. Currently the experiments presented in the appendix are more compelling than the experiments presented in the main paper.
Given elements they provided in the discussion with the reviewers, the authors should also emphasize more clearly in the paper how their proposed architecture differs from other scattering GCNs that have been proposed, and I would suggest to do an ablation study to show that the enhancements that they introduced in the architecture are actually useful.

A consensus between all reviewers could unfortunately not be found:
- Two reviewers were satisfied with the way the authors had addressed their concerns and with the additional experiments proposed.
 - One reviewer considers that the idea of using the scattering transform in this application is not a sufficient contribution to grant publication.

Given that
- two reviewers find the contribution compelling and their concerns are well addressed
-  the use the geometric scattering transform is simple and yet effective both computationally and in terms of the ability of the method to identify larger cliques
- the sole motivation of the reviewer who votes for rejection is a claim that the scientific contribution is not sufficient against the opinion of the two reviewers and that of the AC,

the AC is in favor of acceptance.

### Acknowledging that the proposed loss function is the same as in Karalias and Loukas (2021) !

One element which is very important is that the discussion with one of the reviewers has clearly established that **the loss function introduced in this paper is exactly the same** (up to a constant and a multiplicative factor) **as the loss function $\ell_{\text{clique}}$ obtained in** Corollary 1 of **Karalias and Loukas (2021)**.

In the discussion with the reviewer, the authors wrote
 "We are happy to add discussion and clarification of the loss terms to our manuscript. This discussion and clarification can also help readers to understand the model better." (which I entirely agree with) but they did not act upon that, yet...

It would now be more than **absolutely necessary to add that discussion** ! This will add value to the paper as it will show that the proposed loss is less ad hoc than it might seem, given that it can be obtained via at least two routes. Moreover establishing connections between approaches in the literature is clearly a valuable contribution.

Currently, the conclusion says: "We further construct a two-term loss function which [...]" which still strongly suggests that the loss function is novel, and it therefore very problematic ethically. The sentence added in blue on line 186 is not sufficient to address the issue.

**The authors should** at the very least **add a sentence** at the beginning of section 3.4 **saying** something like: "We propose a simple derivation of a multi-objective loss function, and retrieve **a loss function which was also obtained by Karalias and Loukas (2021)** as a natural upper bound to the probabilistic penalty loss that they propose".
And at the end of section 3.4, the authors should add a sentence saying: **"The proposed loss matches the loss $\ell_{\text{clique}}$ obtained in Corollary 1, Section 4.1 of Karalias and Loukas (2021)."**

**Award:**

No

---

### Decision · Program_Chairs · 2022-09-14

Accept